# Organoids: A New Chapter in Sarcoma Diagnosis and Treatment

**DOI:** 10.3390/ijms231911271

**Published:** 2022-09-24

**Authors:** Iason Psilopatis, Stefania Kokkali, Kostas Palamaris, Antonia Digklia, Kleio Vrettou, Stamatios Theocharis

**Affiliations:** 1Department of Gynecology, Charité-Universitätsmedizin Berlin, Corporate Member of Freie Universität Berlin and Humboldt–Universität zu Berlin, Augustenburger Platz 1, 13353 Berlin, Germany; 2First Department of Pathology, Medical School, National and Kapodistrian University of Athens, 75 Mikras Asias Street, Bld 10, Goudi, 11527 Athens, Greece; 3Section of Medical Oncology, 2nd Department of Medicine and Laboratory, National and Kapodistrian University of Athens, Hippocratio General Hospital of Athens, V. Sofias 108, 11526 Athens, Greece; 4Department of Oncology, Lausanne University Hospital and University of Lausanne, CH-1011 Lausanne, Switzerland

**Keywords:** organoids, sarcoma, three-dimensional cell culture

## Abstract

Sarcomas are malignant tumors of mesenchymal origin that can occur at any age. The rarity of these tumors in combination with the vast number of histological subtypes render the study of sarcomas challenging. Organoids represent complex three-dimensional cell culture systems, deriving from stem cells and preserving the capacity to differentiate into the cell types of their tissue of origin. The aim of the present review is to study the current status of patient-derived organoids, as well as their potential to model tumorigenesis and perform drug screenings for sarcomas. In order to identify relevant studies, a literature review was conducted and we were able to identify 16 studies published between 2019 and 2022. The current manuscript represents the first comprehensive review of the literature focusing on the use of organoids for disease modelling and drug sensitivity testing in diverse sarcoma subtypes.

## 1. Introduction

Sarcomas comprise a heterogeneous group of connective tissue malignancies of mesenchymal origin, including numerous distinct diagnostic entities [1]. Despite the large number of subtypes, sarcomas can be divided into soft tissue sarcomas (STS) and Primary bone sarcomas (PBS) [2]. Rhabdomyosarcoma and non-rhabdomyosarcoma STS represent the two major histological classes of STS, whereas osteosarcoma and Ewing sarcoma are the most common histological subtypes of PBS [3]. According to the American Cancer Society, about 13,190 new cases of STS and 3910 cases of primary cancer of the bones and joints will be diagnosed in the United States in 2022, leading to about 5130 and 2100 deaths, respectively [4,5]. STS usually arise in the extremities or the retroperitoneum, and most patients notice an eventually painful growing tumor mass [6]. The 5-year survival rate for people diagnosed with STS drops to 15% when a distant metastasis is present [7]. PBS most commonly cause pain, swelling and pathological bone fractures [8]. Even though most PBS are found at an early stage [9], the 5-year survival rate for patients diagnosed at a distant surveillance, epidemiology, and end results (SEER) stage amounts to 23–39%, depending on the histological subtype [10,11,12]. Diagnostic evaluation of sarcoma includes, in addition to a physical examination, a plain X-ray, computed tomography (CT) scans, and magnetic resonance imaging (MRI), eventually combined with a positron emission tomography (PET) scan. Nevertheless, definite diagnosis always requires biopsy of the tumor mass [13,14]. For patients with early stage resectable sarcoma, surgical excision represents the mainstay of treatment, combined with chemotherapy for chemosensitive histotypes (mainly Ewing sarcoma and osteosarcoma, whereas in STS, it may be administered in certain cases of high-risk tumors). Patients with advanced disease are mainly treated with systemic therapy [15,16].

Organoids represent three-dimensional in vitro growing systems that derive from self-organizing cells, capable of recapitulating the in vivo structural and functional features of an organ [17]. Organoids may originate from embryonic, induced pluripotent, neonatal, or adult stem cells [18]. Their establishment requires scaffold or scaffold-free techniques to avert direct physical contact to the plastic dish [19]. The most commonly used scaffold is Matrigel, a heterogeneous and gelatinous protein mixture purified from Engelbreth–Holm–Swarm mouse sarcoma cells that resembles the natural extracellular matrix (ECM) [20]. The use of Matrigel or similar hydrogels has successfully enabled the generation of gastrointestinal, salivary gland, hepatic, pancreatic, brain, retinal, renal, pulmonary, or gynecological organoids [21]. The application of organoids ranges from human developmental biology, disease modeling, and tissue engineering to regenerative medicine, personalized medicine, and drug screening [22]. Importantly, given that organoids are expandable, cryopreservable, and genetically modifiable, they allow for various applications in cancer research as well, particularly through the development of tumor organoids [23,24].

In this review, we extensively investigated the use of organoids for disease modeling and drug sensitivity testing in sarcoma (Figure 1). The literature review was conducted using the MEDLINE, LIVIVO, and Google Scholar databases. We included only original research articles and scientific abstracts written in the English language, that explicitly reported on the development of three-dimensional sarcoma organoid models. Studies incorporating two-dimensional or spheroid sarcoma models, as well as studies not clearly stating the use of organoid sarcoma models were excluded. The search terms sarcoma, rhabdomyosarcoma, fibrosarcoma, carcinosarcoma, osteosarcoma, chondrosarcoma, and organoid were employed and we were able to identify a total of 586 articles published between 1953 and April 2022, after the exclusion of duplicates. A total of 554 were discarded in the initial selection process after abstract review. The full texts of the remaining 32 publications were evaluated, and after detailed analysis, a total of 16 relevant studies published between 2019 and April 2022, that met the inclusion criteria, were selected for the literature review. Figure 2 presents an overview of the selection process.

## 2. Soft Tissue Sarcoma

### 2.1. Rhabdomyosarcoma

Two study groups have recently described their initial data on rhabdomyosarcoma organoids. Gatzweiler et al. used tumor samples from the INFORM pediatric precision oncology program (individualized therapy for relapsed malignancies in childhood) to study the molecular tumor profile and the drug-screening results of long-term embryonal rhabdomyosarcoma organoid-like cultures, and concluded that these organoids not only preserve the molecular characteristics of the original tumor, but also yield a sufficient amount of viable cells for the evaluation of drug combinations [25]. Meister et al. generated pediatric-patient-derived rhabdomyosarcoma organoid models comprising all major subtypes, and found that rhabdomyosarcoma organoids retain marker protein expression, represent the diverse clinical presentation of the different histopathological subtypes, as well as molecularly resemble the tumor of origin. Moreover, with 5/7 tested lines reaching passage 40, the models remain genetically and transcriptionally stable after culture over time, while rhabdomyosarcoma organoid drug screening reflects established drug sensitivities. Of note, the study group succeeded in genetically editing the organoid models using CRISPR/Cas9 and described that p53-deficient embryonal rhabdomyosarcoma tumoroid cells show a more sensitive response to the checkpoint kinase inhibitor prexasertib [26].

### 2.2. Non-Rhabdomyosarcoma

Several studies have highlighted the usefulness of non-rhabdomyosarcoma organoid models. Gaebler et al. managed to create long-term non-rhabdomyosarcoma organoids deriving from patients with myxoid liposarcoma, undifferentiated pleomorphic sarcoma or biphasic synovial sarcoma. This organoid platform allowed for drug screening of a set of compounds resembling first-line chemotherapy and novel compounds, whereas a multiplexed protein-profiling assay provided an insight into (phospho)-proteomics [27]. Boulay et al. generated patient-derived synovial sarcoma organoids and performed genome-wide epigenomic profiling, with a view to studying specific chromatin-remodeling mechanisms and dependencies. Primary synovial sarcoma organoids were shown to display distinctive patterns of BRG1/BRM-associated factor (BAF) complex distribution, while broad BAF complex domains correlated with active chromatin states, as well as the expression of a tumor-specific gene signature. Additionally, the presence of polycomb repressive complex 1 (PRC1) and the related H2AK119ub histone mark at broad BAF domains was confirmed, and synovial sarcoma cells were more sensitive to incremental doses of the ubiquitin-specific protease 7 (USP7) inhibitor FT827 than Ewing sarcoma cells [28]. Maloney et al. developed patient-derived skin fibrosarcoma organoids, which were subjected to the tyrosine kinase inhibitor imatinib or the antibiotic anthracycline chemotherapy agent doxorubicin. Interestingly, a significant decrease in adenosine 5′-triphosphate (ATP) activity was reported only after the organoid cultures had been subjected to a high concentration of imatinib, whereas low concentrations of doxorubicin generated a significant reduction in ATP activity [29]. Maru et al. investigated the transformation potential of the combination of *Kirsten rat sarcoma virus* (*Kras*) activation and *Phosphatase and tensin homolog* (*Pten*) inactivation in murine endometrial organoids in the subcutis of immunodeficient mice, and reported that *Cyclin-Dependent Kinase Inhibitor 2A* (*CDKN2A*) knockdown or transformation-related protein 53 (*Trp53*) deletion led to the induction of sarcomatous differentiation and, consequently, to the development of uterine carcinosarcoma [30]. The Japanese study group also developed *Trp53* wildtype fallopian tube organoids expressing the mutant *Kras*, that were found to develop ovarian carcinosarcoma upon *CDKN2A* suppression [31]. McCorkle et al. successfully established in vitro monolayer and ovarian carcinosarcoma organoid cell lines to investigate the effects of diverse chemotherapeutic agents. Notably, dose–response curves revealed a relatively high resistance to carboplatin, gemcitabine, and topotecan, but sensitivity to paclitaxel, doxorubicin, and artesunate [32].

Table 1 summarizes the results of the different studies on STS organoids.

## 3. Primary Bone Sarcoma

### 3.1. Osteosarcoma

A handful of studies have investigated the use of organoids for disease modeling and drug sensitivity testing in osteosarcoma. Wang et al. harvested early passage osteosarcoma cells from mice tumors to develop osteosarcoma organoids and proved that the inhibition of p27 degradation by S-Phase Kinase Associated Protein 2 (SKP2) significantly delays osteosarcoma development and progression, induces apoptosis, and diminishes tumor-initiating properties in the organoid model [33]. Subramaniam et al. generated multicell-type lung organoid models with osteosarcoma cells and reported a significant reduction in osteosarcoma cell growth after treatment with pimozide [34]. In her study, Johansson created patient-derived osteosarcoma organoids displaying rounded structure in microscopy images and secreting Vascular Endothelial Growth Factor (VEGF) under the cultivation. By performing cell viability assays on both the organoids and the cryopreserved cancer cells from the original tumor, the author described similar resistance profiles [35]. Last but not least, He et al. firstly generated a patient-derived organoid platform for lung metastatic osteosarcoma that preserved the cellular morphology and expression of the osteosarcoma markers Vimentin and SRY-Box Transcription Factor 9 (Sox9). Interestingly, given that the primary lung metastatic osteosarcoma organoids retained the T-cell distribution of the parental tumors, anti-programmed cell death protein 1 (PD1) treatment was found to activate CD8^+^ T-cells in the organoid cultures [36].

### 3.2. Chondrosarcoma

Only one study group has, to date, reported on the experience with a three-dimensional chondrosarcoma organoid model. Veys et al. created SW1353-cell-derived chondrosarcoma organoids to test the anti-tumor activity of microRNA-342-5p and microRNA-491-5p, and concluded that microRNA-342-5p significantly promotes apoptosis, especially in hypoxia [37].

### 3.3. Ewing Sarcoma

Two study groups have recently described their initial experiences with Ewing sarcoma organoids. Maurer et al. developed Ewing sarcoma organoids and monolayers from a metastatic pulmonary lesion from a patient with an inherited *BRCA1 Associated RING Domain 1* (*BARD1*) mutation. The organoids surprisingly demonstrated high sensitivity to poly (ADP-ribose) polymerase (PARP) inhibitors [38]. Two years after their initial publication, the same study group published the results of their second study, stating that the loss of *BARD1* increases Ewing sarcoma sensitivity to DNA damage, and that Guanylate-binding protein 1 (GBP1) expression contributes to DNA damage response in Ewing sarcoma organoids [39]. Komatsu et al., on the other hand, were the first to use patient-derived cell lines of *CIC-DUX4* sarcoma to generate Ewing-like small round cell sarcoma organoids. Notably, drug sensitivity assays revealed a dose-dependent decrease in organoid size after treatment with two different concentrations of gemcitabine [40].

Table 2 summarizes the results of the different studies on PBS organoids.

## 4. Conclusions

Organoids, in general, offer a unique opportunity, since they are translatable, reproducible, and scalable. Their generation from pluripotent or adult stem cells renders them an exceptional three-dimensional culture system, capable of closely mimicking the architecture and physiology of the tissue of origin [41]. Nonetheless, organoids show a relatively random growth nature, do not support interorgan communication, and lack vasculature and immune cells [42]. As such, the development of standardized protocols for routine organoid handling is of utmost importance, in order to increase the success rate of organoid generation and research [43].

Sarcoma is a challenging disease, and the lack of reliable treatments in this field renders paramount the development of new in vivo assays that can support drug discovery. However, for the time being, there has been limited success in cell culture, and great difficulty in identifying pure sarcoma organoids. Formulation of optimal culture media and maintenance processes has been difficult, as it depends on each subtype, yet sarcomas, unlike epithelial-based tumors, are notorious for their heterogeneity, defined as the existence of distinct cell subpopulations with varying inter- and intra-tumor morphological, genotypic, and phenotypic features [44].

One other challenge to overcome is the limited available amount of tumor tissue to be used for organoid culture. It has been shown that increasing the amount of starting material, e.g., by taking multiple biopsies from epithelial-based tumors, improves the outgrowth of tumor organoids. In sarcoma, however, the risk of contamination to other tissues during biopsy and the large quantity of tissue required for complex diagnosis render the available material from biopsies to be used for organoids often restricted. Consequently, these limitations also hinder the study of inter- and intra-patient heterogeneity in sarcomas.

Several literature reviews have been recently published on three-dimensional sarcoma models [45,46,47]. These publications mostly report on sarcoma-derived three-dimensional cultures or spheroids and only underline the importance of the generation of advanced sarcoma organoids. Notably, both Kapalczynska et al. and Jensen et al. recently stated that, compared to two-dimensional cell cultures, three-dimensional models mimic the in vivo cell environment, and may, thus, offer greater opportunities to study cellular signaling or drug sensitivity [48,49]. Moreover, Gunti et al. compared spheroids with organoids, and concluded that, although spheroid establishment is simpler, organoids represent long-term, cryopreservable culture models, histologically and genetically resembling the tissue of origin [50]. A major advantage of 3D systems such as organoids is the preservation of intra-tumor heterogeneity, that allows them to recapitulate human tumors much more closely, compared to their more conventional 2D cell culture methods. Carcinogenesis is a multi-step process that evolves through the progressive emergence of neoplastic clones, characterized by distinct genetic and epigenetic alteration. Malignant tumors are, therefore, collections of molecularly divergent cell populations which, in conditions of selection pressure, such as those encountered during treatment, provide the fuel for therapy resistance. Thus, the potential of organoids to capture the heterogeneity of original tumors makes them an ideal tool for the evaluation of novel drugs and the clarification of drug-resistance mechanisms.

To our knowledge, this is the first comprehensive literature review focusing on the use of organoids for disease modeling and drug sensitivity testing in different sarcoma subtypes. We were able to identify sixteen original articles on the successful development of mostly patient-derived sarcoma organoids, with a total of six studies presenting novel results in terms of non-rhabdomyosarcoma pathogenesis and/or treatment [27,28,29,30,31,32], followed by four manuscripts providing valuable information on osteosarcoma treatment options [33,34,35,36]. These organoid cultures represented useful preclinical models for the identification of molecular mechanisms correlating with sarcoma genesis and progression, and, most importantly, allowed for drug screening assays, thus paving the way for the establishment of novel potent treatment tools. Given that STSs alone are further subdivided into approximately 70, partly extremely rare, morphologically distinct subtypes [51], the development of patient-derived sarcoma organoids undoubtedly marks the beginning of a new era in sarcoma diagnosis and treatment. Of note, future studies should not put their focus only on main sarcoma subtypes, but also try to establish rare sarcoma organoids, in order to shed light on the countless questions of both involved clinicians and affected patients. All in all, the establishment of next-generation organoids requires a reduction in high organoid diversity, organoid maturation promotion, generation of larger organoids, as well as high-throughput live imaging [52].

## Figures and Tables

**Figure 1 ijms-23-11271-f001:**
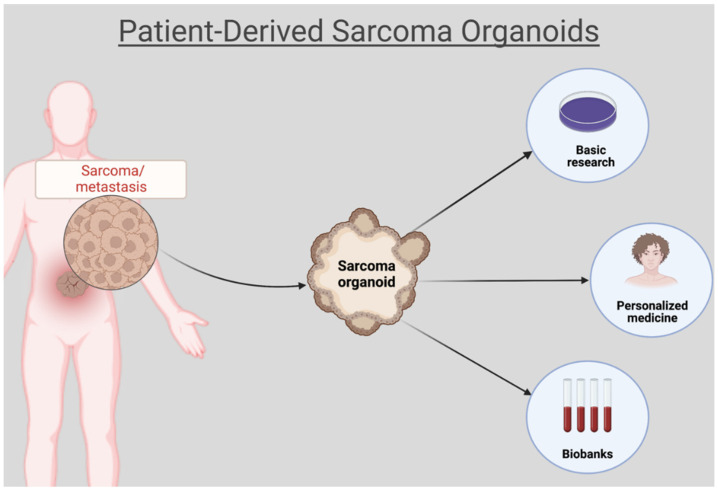
Patient-derived sarcoma organoids for disease modeling and drug sensitivity testing. Created with BioRender.com (accessed on 26 July 2022).

**Figure 2 ijms-23-11271-f002:**
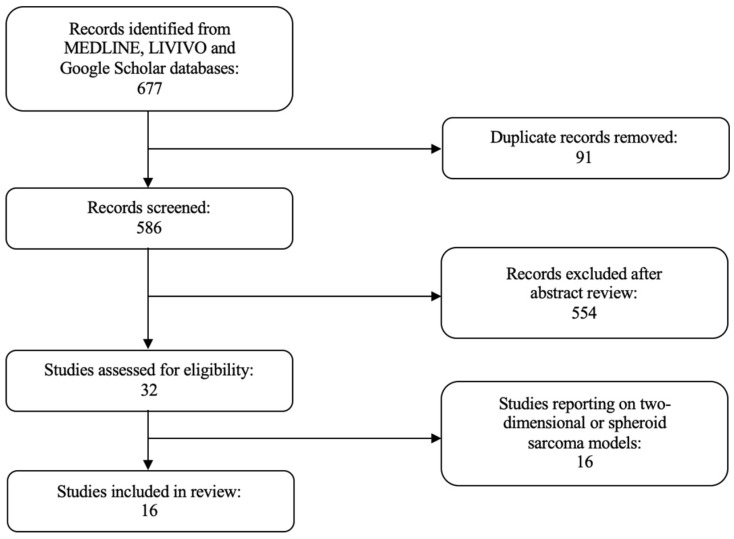
PRISMA flow diagram visually summarizing the screening process.

**Table 1 ijms-23-11271-t001:** Use of organoids in different types of soft tissue sarcoma.

STS Type	Organoids	Culture Conditions	Main Results	References
Embryonal rhabdomyosarcoma	Patient-derived long-term organoid-like cultures	Fresh patients’ surgical specimens or mouse patient derived xenografts dissociated mechanically and enzymatically (collagenase and trypsin) Sub-culture of free-floating semi-adherent spheroids by dissociation with TrypLE Seeding in a 1:2 to 1:5 ratio in fresh TSM complete medium, containing antibiotics and growth factors (EGF, FGF, PDGF) Six-times passaged cultures	Preservation of the main molecular characteristics of the embryonal rhabdomyosarcoma Shift in mean drug sensitivity Zebrafish embryonal rhabdomyosarcoma model showed most sensitive response to idasanutlin and navitoclax	[25]
Rhabdomyosarcoma	Patient-derived rhabdomyosarcoma organoids	Needle biopsies, surgical specimens or bone marrow aspirates Minced pieces in extracellular matrix or single-cell suspensions in extracellular matrix (ECM) substitute Basement-membrane-extract-supplemented medium, that contained antibiotics and growth factors (EGF, FGF, and IGF) Tumor cell outgrowth to two-dimensional monolayers Further propagation and expansion	Preservation of original tumor characteristics and clinical presentation Genetic and transcriptional stability of tumoroid models over time Drug screening reflects established drug sensitivities, with higher sensitivity of p53 deficient organoid cells to prexasertib	[26]
Myxoid liposarcoma, undifferentiated pleomorphic sarcoma, biphasic synovial sarcoma	Patient-derived non-rhabdomyosarcoma organoids	Sample dissociation Seeding into 24w plates in matrix-like scaffolds Growth until >100 μm and harvesting	Pharmacokinetic properties profiling through high-throughput drug screening (Phospho)-proteomics analysis via multiplexed protein-profiling assay	[27]
Synovial sarcoma	Synovial sarcoma and Ewing sarcoma patient-derived tumor organoids	Fresh patients’ surgical specimens dissociated and cultured in IMDM Supplementation with 20% KO serum, 10 ng/mL human recombinant EGF and basic fibroblast growth factor, and 1% Pen/Strep in ultra-low attachment flasks	Correlation of broad BAF complex domains with active chromatin states and the expression of a tumor-specific gene signature Reversible BAF complex retargeting through SS18-SSX expression and consequent functional PRC1–PRC2 complex uncoupling USP7 depletion as an epigenetic vulnerability in synovial sarcoma	[28]
Skin fibrosarcoma	Patient-derived skin fibrosarcoma organoids	Immersion bioprinting of collagen–hyaluronic acid bioinks in 96-well plates Mechanical and enzymatic (collagenase, protease, and hyaluronidase) dissociation of human surgical specimens Maintenance in DMEM-HG supplemented with 10% FBS, 1% L-glutamine and 1% penicillin/streptomycin with 5% CO_2_	Successful employment of bioprinted patient-derived sarcoma organoids for chemotherapy screening studies Significant decrease in ATP activity after imatinib and doxorubicin increase	[29]
Uterine carcinosarcoma	Murine endometrial organoids	Mechanical and enzymatic (collagenase, dispase II) dissociation of mouse fresh tissues Resuspended in solidified Matrigel and cultured in medium supplemented with R-spondin1, Noggin, Jagged-1, Y27632) (MBOC: Matrigel bilayer organoid culture protocol)	Carcinosarcoma development in *Kras^G12D^* organoids with *Cdkn2a* knockdown or *Trp53* deletion Epithelial–mesenchymal transition (EMT) state preservation, as well as presentation of heterogeneous statuses in the *Kras* loci in tumor-derived organoids	[30]
Ovarian carcinosarcoma	Murine fallopian tube organoids	Dissociation of fresh mouse tissues into single cells, which are then suspended in Matrigel (MBOC: Matrigel bilayer organoid culture protocol)	Cooperation of Kras activation with p53 loss in terms of carcinosarcoma genesis Resistance of *Kras^G12D^*-driven carcinosarcoma-derived organoids to paclitaxel and cisplatin	[31]
Ovarian carcinosarcoma	Ovarian carcinosarcoma organoid cell lines	Tumor tissue harvesting Digestion with highly purified collagenase I and II	High resistance to carboplatin, gemcitabine, and topotecan Sensitivity to paclitaxel, doxorubicin, and artesunate	[32]

**Table 2 ijms-23-11271-t002:** Use of organoids in different types of primary bone sarcoma.

PBS Type	Organoids	Culture Conditions	Main Results	References
Osteosarcoma	Osteosarcoma organoid culture	Not specified	Slower proliferation of Osx1-Cre; Rb1^lox/lox^; Trp53^lox/lox^; p27^T187A/T187A^ tumors both in vivo and in vitro organoid C1 and Pevenodistat showed selective inhibition in Osx1-Cre; Rb1^lox/lox^; Trp53^lox/lox^ in osteosarcoma organoid	[33]
Osteosarcoma	Multicell-type lung organoid model	Cell mix with KHOS/NP GFP positive cells Growth in an ultra-low attachment using specific spheroid media	Significant growth reduction in osteosarcoma cells after treatment with 20 μM pimozide	[34]
Osteosarcoma	Patient-derived osteosarcoma organoids	Primary cell-line-derived osteosarcoma patient Single-cell suspensions in Basement-Menbrane Extract-supplemented medium, with the addition of FGF and EGF 6-well plate incubation Cell expansion and cryopreservation	Gradual increase in VEGF level in osteosarcoma organoids Same tendency of survival index toward chemotherapy, yet different resistance toward oxaliplatin compared to cryopreserved and original cancer cells	[35]
Osteosarcoma	Patient-derived lung metastatic osteosarcoma organoids	Surgical specimens of primary or metastatic osteosarcomas Cut/EnBloc protocol Single-cell model	Histological feature and T-cell distribution maintenance of parental osteosarcoma lung metastatic tumors CD8^+^ T-cell activation through PD-1 immune checkpoint blockade	[36]
Chondrosarcoma	Three- dimensional chondrosarcoma organoid model	Chondrosarcoma cell line Cell growth in collagen I/III scaffolds Seeding in 96-well culture plates Incubation at 5% CO_2_ in HG-DMEM supplemented with 10% FCS and antibiotics Transfer to 24-well plates Incubation in the same medium pre-equilibrated with 3% O_2_	Cell death induction via microRNA-342-5p on a three-dimensional chondrosarcoma organoid model under hypoxia	[37]
Ewing sarcoma	Patient-derived Ewing-sarcoma organoids	Not specified	High sensitivity to PARP inhibitors Sensitization of PARP inhibitor-resistant Ewing cell lines to PARP inhibition via BARD1 small-interfering-RNA Upregulation of tumor cell surface expression of PD-L1 after PARP inhibition Early tumor cell death after PD-1 blocking antibody addition to T-cell/tumor cell cocultures	[38]
Ewing sarcoma	Patient-derived Ewing-sarcoma organoids	Embedding of tumor pieces in growth-factor-reduced Matrigel Incubation at 5% CO_2_, with media exchange every 3 days	Upregulation of immunoregulatory pathways upon relapse High Ewing sarcoma sensitivity to DNA damage, talazoparib treatment and radiation after *BARD1* loss GBP1 expression promotes DNA damage response in Ewing sarcoma organoid cells	[39]
Ewing-like small round cell sarcoma	Tumor organoids from chorioallantoic membrane tumor	Enzymatic digestion with Trypsin-EDTA and Liberase Cell inoculation to a 96-well plate Incubation at CO_2_	Tumor organoid formation from the chorioallantoic membrane tumor *CIC-DUX4* gene retention in chorioallantoic membrane tumor organoids Dose-dependent effect of gemcitabine on organoid size	[40]

## Data Availability

Not applicable.

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
