# Peer review of "Organoids: A New Chapter in Sarcoma Diagnosis and Treatment"

_ijms, 2022, doi:10.3390/ijms231911271_

Round 1
Reviewer 1 Report
This review by Psilopatis et al. titled “Organoids: A New Chapter in Sarcoma Diagnosis and Treatment” is a literature review of 16 studies published between 2019 and 2022 regarding the application of organoid technology to the study of sarcomas, malignant tumors of mesenchymal origin that remains relatively understudied compared to carcinomas. The use of organoids is rapidly advancing both the basic and translational aspects of oncology research, and I felt this review is a timely one. However, I think the manuscript could be improved as I detail below.
1, 16 studies seem somewhat a small number to perform a literature review. If the number of studies could be increased if the time window (2019-2022) is widened, please consider doing so. (A related question: how was this time window selected?)
2, Criteria for paper selection should be explicitly stated (e.g., what formula was used to search in what database, whether/how many/why studies were excluded, etc.).
3, Please explain in more detail why the use of organoids should be considered a better alternative to conventional 2D culture. If there are studies that compare the two (for example in terms of cellular signaling or drug sensitivity), it would be nice to know the experimental evidence.
4, The authors mention in the conclusion section that optimization of culture conditions is difficult for organoid culture, especially for sarcoma which is a highly heterogeneous disease entity. It would thus be a great asset to researchers aspiring to enter this field if the authors could perhaps create a table that outlines the various reported organoid culture conditions (and to point the readers to the original paper that established the conditions).
Author Response
Reviewer 1
This review by Psilopatis et al. titled “Organoids: A New Chapter in Sarcoma Diagnosis and Treatment” is a literature review of 16 studies published between 2019 and 2022 regarding the application of organoid technology to the study of sarcomas, malignant tumors of mesenchymal origin that remains relatively understudied compared to carcinomas. The use of organoids is rapidly advancing both the basic and translational aspects of oncology research, and I felt this review is a timely one. However, I think the manuscript could be improved as I detail below.
1) 16 studies seem somewhat a small number to perform a literature review. If the number of studies could be increased if the time window (2019-2022) is widened, please consider doing so. (A related question: how was this time window selected?)
We would like to thank the reviewer for this suggestion. Indeed, 16 studies is a relatively small number to perform a literature review. However, we meticulously searched the MEDLINE, LIVIVO and Google Scholar databases for relevant articles without restricting the time window, but were able to find only the reported sixteen articles which were published between 2019 and April 2022. To our knowledge, the present review is the most comprehensive literature review on the use of organoids for disease modeling and drug sensitivity testing in sarcoma. To avoid causing confusion to the readers, we added a paragraph stating that:
“The search terms sarcoma, rhabdomyosarcoma, fibrosarcoma, carcinosarcoma, osteosarcoma, chondrosarcoma, and organoid were employed and we were able to identify a total of 586 articles published between 1953 and April 2022, after exclusion of duplicates. 554 were discarded in the initial selection process after abstract review. The full texts of the remaining 32 publications were evaluated, and after detailed analysis, a total of 16 relevant studies published between 2019 and April 2022, that met the inclusion criteria, were selected for the literature review.”
2) Criteria for paper selection should be explicitly stated (e.g., what formula was used to search in what database, whether/how many/why studies were excluded, etc.).\
Thank you for this useful comment. We have now added a paragraph explicitly stating the inclusion/exclusion criteria for paper selection, as well as a flow chart depicting the selection process:
“We included only original research articles and scientific abstracts written in the English language, that explicitly reported on the development of three-dimensional sarcoma organoid models. Studies incorporating two-dimensional or spheroid sarcoma models, as well as studies not clearly stating the use of organoid sarcoma models were excluded. The search terms sarcoma, rhabdomyosarcoma, fibrosarcoma, carcinosarcoma, osteosarcoma, chondrosarcoma, and organoid were employed and we were able to identify a total of 586 articles published between 1953 and April 2022, after exclusion of duplicates. 554 were discarded in the initial selection process after abstract review. The full texts of the remaining 32 publications were evaluated, and after detailed analysis, a total of 16 relevant studies published between 2019 and April 2022, that met the inclusion criteria, were selected for the literature review. Figure 2 presents an overview of the selection process.”
3) Please explain in more detail why the use of organoids should be considered a better alternative to conventional 2D culture. If there are studies that compare the two (for example in terms of cellular signaling or drug sensitivity), it would be nice to know the experimental evidence.
We would like to thank the reviewer for this very helpful suggestion. We have now added a paragraph explaining why the use of organoids should be considered a better alternative to conventional 2D culture, based on studies comparing the two models:
“Notably, both Kapalczynska et al. and Jensen et al. recently stated that, compared to two-dimensional cell cultures, three-dimensional models mimic the in vivo cell environment, and may, thus, offer greater opportunities to study cellular signaling or drug sensitivity49,50. Moreover, Gunti et al. compared spheroids with organoids, and concluded that, although spheroid establishment is simpler, organoids represent long-term, cryopreservable culture models, histologically and genetically resembling the tissue of origin51. A major advantage of 3D systems, such as organoids is the preservation of intra-tumor heterogeneity, that allows them to recapitulate human tumors much more closely, compared to their more conventional 2D cell culture methods. Carcinogenesis is a multi-step process, that evolves through the progressive emergence of neoplastic clones, characterized by distinct genetic and epigenetic alteration. Malignant tumors are, therefore, collections of molecularly divergent cell populations which, in conditions of selection pressure, such as these encountered during treatment, provide the fuel for therapy resistance. Thus, the potential of organoids to capture heterogeneity of original tumors makes them an ideal tool for the evaluation of novel drugs and the clarification of drug-resistance mechanisms.’’
4) The authors mention in the conclusion section that optimization of culture conditions is difficult for organoid culture, especially for sarcoma which is a highly heterogeneous disease entity. It would thus be a great asset to researchers aspiring to enter this field if the authors could perhaps create a table that outlines the various reported organoid culture conditions (and to point the readers to the original paper that established the conditions).
Thank you for this productive comment. We have now added two new columns in tables 1 and 2 outlining the various organoid culture conditions and pointing the readers to the original papers that established the conditions.
Reviewer 2 Report
The diagnosis and treatment of sarcoma is a challenge and personalized cancer treatment is necessary. We require new models that can reliably recapitulate both intra- and inter-tumor heterogeneity.This paper gives a good overview of the organoids within sarcoma, however, there are a few points that needs to be addressed.
1.Tumor heterogeneity is one of the most important issues for tumor research, and the authors may need give a clear explanation of its definition and association with organoids.
2.Many studies of organoids focus on epithelial-based tumors. Compared with epithelial-based tumors, the problems and unique features with sarcoma organoids is necessary to explain.
3.The discussion part is to sample ,and the limitations of current techniques, advancements in standardization for organoids and future opportunities for organoid generation need more explanation.
4. The only one figure, Figure 1 is overly simplistic.
Author Response
Reviewer 2
The diagnosis and treatment of sarcoma is a challenge and personalized cancer treatment is necessary. We require new models that can reliably recapitulate both intra- and inter-tumor heterogeneity. This paper gives a good overview of the organoids within sarcoma, however, there are a few points that needs to be addressed.
1) Tumor heterogeneity is one of the most important issues for tumor research, and the authors may need give a clear explanation of its definition and association with organoids.
We would like to thank the reviewer for this useful suggestion. We have now added the definition of tumor heterogeneity and its association with organoids:
“Formulation of optimal culture media and maintenance processes has been difficult, as it depends on each subtype, yet sarcomas, unlike epithelial-based tumors, are notorious for their heterogeneity, defined as the existence of distinct cell subpopulations with varying inter- and intra-tumor morphological, genotypic, and phenotypic features45.’’
2) Many studies of organoids focus on epithelial-based tumors. Compared with epithelial-based tumors, the problems and unique features with sarcoma organoids is necessary to explain.
We agree with the reviewer’s point of view, and have now pointed out the problems and unique features with sarcoma organoids, compared with epithelial-based tumors:
“Formulation of optimal culture media and maintenance processes has been difficult, as it depends on each subtype, yet sarcomas, unlike epithelial-based tumors, are notorious for their heterogeneity, defined as the existence of distinct cell subpopulations with varying inter- and intra-tumor morphological, genotypic, and phenotypic features45.”
“It has been shown that increasing the amount of starting material, e.g. by taking multiple biopsies from epithelial-based tumors, improves the outgrowth of tumor organoids.”
3) The discussion part is to sample and the limitations of current techniques, advancements in standardization for organoids and future opportunities for organoid generation need more explanation.
Thank you for this constructive comment. In the revised form of the manuscript, we have expanded the discussion and included sections on the limitations of current techniques, advancements in standardization for organoids, and future opportunities for organoid generation:
“Their generation from pluripotent or adult stem cells renders them an exceptional three-dimensional culture system, capable of closely mimicking the architecture and physiology of the tissue of origin42. Nonetheless, organoids show a relatively random growth nature, do not support interorgan communication, and lack vasculature and immune cells43. As such, the development of standardized protocols for routine organoid handling is of utmost importance, in order to increase the success rate of organoid generation and research44.”
“Formulation of optimal culture media and maintenance processes has been difficult, as it depends on each subtype, yet sarcomas, unlike epithelial-based tumors, are notorious for their heterogeneity, defined as the existence of distinct cell subpopulations with varying inter- and intra-tumor morphological, genotypic, and phenotypic features45.”
“Notably, both Kapalczynska et al. and Jensen et al. recently stated that, compared to two-dimensional cell cultures, three-dimensional models mimic the in vivo cell environment, and may, thus, offer greater opportunities to study cellular signaling or drug sensitivity49,50. Moreover, Gunti et al. compared spheroids with organoids, and concluded that, although spheroid establishment is simpler, organoids represent long-term, cryopreservable culture models, histologically and genetically resembling the tissue of origin51. A major advantage of 3D systems, such as organoids is the preservation of intra-tumor heterogeneity, that allows them to recapitulate human tumors much more closely, compared to their more conventional 2D cell culture methods. Carcinogenesis is a multi-step process, that evolves through the progressive emergence of neoplastic clones, characterized by distinct genetic and epigenetic alteration. Malignant tumors are, therefore, collections of molecularly divergent cell populations which, in conditions of selection pressure, such as these encountered during treatment, provide the fuel for therapy resistance. Thus, the potential of organoids to capture heterogeneity of original tumors makes them an ideal tool for the evaluation of novel drugs and the clarification of drug-resistance mechanisms.’’
“All in all, the establishment of next generation organoids requires reduction of high organoid diversity, organoid maturation promotion, generation of larger organoids, as well as high-throughput live imaging53.”
4) The only one figure, Figure 1 is overly simplistic.
Given the limited number of available studies on the use of organoids for disease modeling and drug sensitivity testing in sarcoma, and the sparse initial results on the different sarcoma subtypes, we decided to create a relatively simple figure introducing the reader to this still ‘’obscure’’ topic, and symbolizing our only primitive experience with the use of sarcoma organoids.
Round 2
Reviewer 1 Report
Thank you for the revisions. The authors have addressed the comments I have raised in my previous review. I would just like to kindly suggest tidying Figure 2 and Tables 1 & 2 if possible, but otherwise I think the review will prove useful to those interested in using organoids for sarcoma research.
Author Response
Thank you for the revisions. The authors have addressed the comments I have raised in my previous review. I would just like to kindly suggest tidying Figure 2 and Tables 1 & 2 if possible, but otherwise I think the review will prove useful to those interested in using organoids for sarcoma research.
We would like to thank the reviewer for this useful comment. We have now tidied Figure 2 and Tables 1 & 2 as possible.